# Gene Expression Profiling of Skeletal Muscles

**DOI:** 10.3390/genes12111718

**Published:** 2021-10-28

**Authors:** Sarah I. Alto, Chih-Ning Chang, Kevin Brown, Chrissa Kioussi, Theresa M. Filtz

**Affiliations:** 1Department of Pharmaceutical Sciences, College of Pharmacy, Oregon State University, Corvallis, OR 97331, USA; altos@oregonstate.edu (S.I.A.); vera0chang@gmail.com (C.-N.C.); Kevin.Brown@oregonstate.edu (K.B.); Theresa.Filtz@oregonstate.edu (T.M.F.); 2Molecular and Cellular Biology Graduate Program, Graduate School, Oregon State University, Corvallis, OR 97331, USA; 3School of Chemical, Biological, and Environmental Engineering, College of Engineering, Oregon State University, Corvallis, OR 97331, USA

**Keywords:** mouse, skeletal muscle, soleus, tibialis anterior, gene expression, metabolism, contraction, signaling

## Abstract

Next-generation sequencing provides an opportunity for an in-depth biocomputational analysis to identify gene expression patterns between soleus and tibialis anterior, two well-characterized skeletal muscles, and analyze their gene expression profiling. RNA read counts were analyzed for differential gene expression using the R package edgeR. Differentially expressed genes were filtered using a false discovery rate of less than 0.05 c, a fold-change value of more than twenty, and an association with overrepresented pathways based on the Reactome pathway over-representation analysis tool. Most of the differentially expressed genes associated with soleus are coded for components of lipid metabolism and unique contractile elements. Differentially expressed genes associated with tibialis anterior encoded mostly for glucose and glycogen metabolic pathway regulatory enzymes and calcium-sensitive contractile components. These gene expression distinctions partly explain the genetic basis for skeletal muscle specialization, and they may help to explain skeletal muscle susceptibility to disease and drugs and further refine tissue engineering approaches.

## 1. Introduction

Skeletal muscles (SM) provide voluntary body movement and locomotion, posture and body position, energy production, fatty acid (FA) β-oxidation, carbohydrate metabolism, and soft tissue support. Skeletal muscles are composed of myofibers, which are formed by fused myoblasts. Myofibers have defining metabolic and contractile properties.

SM contribute to glucose, lipid, and protein metabolism. SM of healthy and well-fed mammals rely on carbohydrate or glucose metabolism as their primary sources of energy from aerobic respiration, anaerobic respiration, or both. SM can also use FAs as a fuel source via β-oxidation when mammals consume a high-fat diet or are in a starving state. As lipid resources become depleted, SM proteins can be broken down for energy use. Oxidative myofibers utilize mostly aerobic respiration. Glycolytic myofibers use primarily anaerobic respiration. Myofibers that utilize both are called oxidative-glycolytic.

Based on contractile properties, myofibers are defined by which myosin heavy chain is present. Myosin heavy chains are a significant component of the thick filament in a sarcomere. Myosin heavy chains (MyHCs) differentiate myofibers into types I (MYH7), II(A) (MYH2), II(B) (MYH4), and II(D or X) (MYH1) [1]. The four contractile properties relate to the three metabolic distinctions. Type I myofibers are oxidative [2]. Type II(A) myofibers are oxidative-glycolytic [2]. Types II(B) and II(D) myofibers are glycolytic [2]. Type II(D) myofibers combine with type II(A) or (B) to create hybrids called II(AD) and II(DB).

The hindlimb SM Soleus (So) and Tibialis Anterior (Ta) were used to determine the gene expression profile of oxidative and glycolytic myofibers by RNA-Seq in one-month-old mice (postpartum day 30, P30), the age at which isolation of single myofibers was feasible. So and Ta act in opposition, wherein one muscle contracts while the other relaxes. So is one of the superficial muscles that shape the lower hindlimb’s posterior component and is essential for ankle joint movement to plantarflex the foot and maintain the standing posture [3]. The murine So is composed of approximately 50% type II(A), 40% type I, and 10% type II(D) myofibers [4]. Ta is located along the anterior lateral side of the tibia; the muscle acts in the foot’s dorsiflexion and inversion, stabilizes the ankle when the foot hits the ground, and pulls the foot off the ground [3]. The murine Ta is composed of approximately 50–60% type II(B), 30–35% type II(D), and 5–10% II(A) myofibers [4].

Pathway analysis identified molecular players involved in glucose and glycogen metabolism, ion transport, and contraction as being overrepresented among the differentially expressed (DE) genes between the So and Ta biopsies. Additionally, DE transcripts with a high fold change (FC) were identified as being involved in neuromuscular junctions and the immune system. Data from this study broadens our understanding of molecular diversity in SM and potentially differential susceptibility to myopathies.

## 2. Materials and Methods

### 2.1. Mice

All animal experiments were performed following the Institutional and National Health and Medical Research Council guidelines. The experimental protocol was approved by the Institutional Animal Care and Use Committee at Oregon State University. Biopsies from *Pitx2^FL/Z^* mice were utilized as previously described for this study [5].

### 2.2. RNA-Sequencing (RNA-Seq) Data Analysis

Five So and two Ta myofiber biopsies were collected at P30. RNA was extracted, sequenced, and analyzed as previously described [5]. The raw RNA-Seq data files for these samples are publicly available through the National Center for Biotechnology Information (NCBI) Sequence Read Archive (SRA) under the reference numbers SRP127367 (So) and SRP145066 (Ta). The reads were aligned using the mouse genome (mm10) and TopHat2. The genomic feature counts were then obtained using HTSeq [5], resulting in a total of 24,421 elements identified. The read counts were normalized to reduce transcript length bias [6]. Longer transcripts appear more prevalent and overshadow smaller transcripts, if not corrected before performing differential expression analysis. Quantitative changes in gene expression levels between the So and Ta groups were discovered using the quasi-likelihood F test from the R package edgeR [7], which resulted in 6123 differentially expressed (DE) genes (*p* < 0.05 c, False Discovery Rate (FDR)-corrected) between So and Ta. The “c” in 0.05 c refers to the *p*-value being FDR-corrected in all instances. A smaller set of DE genes based on a log-fold change of ±1.0 or greater was used, resulting in 2481 DE genes that passed our fold change cutoff.

## 3. Results

### 3.1. RNA-Seq Data Quality Check

So and Ta were processed and subject to RNA-Seq as described above. Principal component analysis of standardized samples showed a clear separation of So and Ta based on RNA expression (Figure 1a). We use red font to denote increased expression in So relative to Ta and blue for decreased expression in So relative to Ta. A volcano plot (negative log of the gene’s FDR-corrected *p*-value vs. log_2_ fold change) of all 24,421 annotated genes showed that the subset of 2481 genes included many highly significant DE genes with increased expression differences in So and Ta (Figure 1b). A heatmap of sample-standardized RNA counts of the final subset of 2481 genes shows clear clustering of myofibers from So and Ta based on gene expression (Figure 1c), as we expect from the biplot in Figure 1a.

Molecular pathways enriched for DE genes from the So and Ta samples were identified using the Reactome database [8]. The 2481 DE genes were matched to molecular pathways, and those pathways were statistically analyzed for over- or under-representation using the Reactome database and its corresponding ReactomePA R package [8,9]. The number of genes analyzed by the Reactome R package decreased to 2198 after removing genes with no Entrez ID. Hypergeometric testing, provided by the ReactomePA R package, identified 19 Reactome pathways as overrepresented with an adjusted *p*-value of less than 0.05 c. Three pathways had fewer than ten associated genes and were removed from further analysis. The DE genes related to the 16 Reactome pathways were ranked from highest to lowest log_2_(FC) values (Appendix A). The ranking revealed that the 16 Reactome pathways could be consolidated into broader categories, such as contraction, ion and amino acid transport, and glucose and glycogen metabolism (Appendix A). We identified 119 DE genes with FC ≥ 20, of which 14 were on the Reactome-curated list (Figure 1d). Examination of known gene functions of the remaining 105 high FC genes revealed that 36 could be manually categorized into lipid metabolism, glycogen metabolism, glucose metabolism, or contraction.

### 3.2. Lipid Metabolism

SM primarily metabolize glucose. Oxidative muscles can also use FAs along with glucose and glycogen as fuel sources. Free FA molecules are transported from the liver via cholesterol transport, which shuttles cholesterol, triglycerides, and protein-bound FAs. Three cholesterol-related genes, *Pon1*, *Tspo2*, and *Apoa2*, were expressed in So by 68, 51, and 27-fold, respectively (Figure 2, Table 1). *Pon1* protects against lipid oxidation for high-density lipoprotein (HDL) cholesterol and low-density lipoprotein (LDL) cholesterol [10]. *Tspo2* traffics free cholesterol in erythroid cells [11]. *Apoa2* regulates steroid concentrations by modulating cholesterol transport [12], the precursor of steroid hormones.

Once free FA molecules pass through the plasma membrane, they enter the mitochondrial matrix to be broken down through β-oxidation. Specific protein channels handle different FA aliphatic tail lengths ranging from greater than 22 to fewer than 5 carbons. *Acsm5* was expressed 30-fold higher in So (Figure 2, Table 1). *Acsm5* encodes for an enzyme that catalyzes the activation of FAs with aliphatic tails of 6 to 12 carbons by CoA to produce acyl-CoA, the first step in FA metabolism [13]. *Dio1*, expressed 69-fold higher in So (Figure 2, Table 1), is involved in FA oxidation and oxidative phosphorylation uncoupling primarily in the liver, kidney, and thyroid [14], in addition to altering the thyroid hormone balance of triiodothyronine (T_3_) and thyroxine (T_4_) [15]. Thyroid hormone signaling regulates numerous genes involved in skeletal muscle homeostasis, function, and metabolism [16].

Several of the highest FC DE genes in So were involved in lipid, sphingolipid, and derived lipid biosynthesis. FA molecules can be elongated and modified into complex and derived lipid molecules. *Tecrl*, involved in FA elongation in polyunsaturated FA biosynthesis [17] and myoblast differentiation upon TNF activation [18], was expressed 55-fold higher in So (Figure 2, Table 1). The increased expression of *Tecrl* could indicate that the So muscle is still developing in P30 mice.

One of the highest FC genes in this study was *Cyp4f39* with a 269-fold higher expression in Ta, which encodes for a FA ω-hydroxylase involved in acyl-ceramide synthesis [19] (Figure 2, Table 1). *Cyp4f39* is involved in cell surface protection, cell recognition, signaling, membrane transportation, increased rigidity, hydrolyzation of very long-chain FAs, and steroid hormone synthesis [19]. Ceramides have been linked to insulin resistance in type II diabetes [20].

### 3.3. Glycogen Metabolism

Glycogen metabolism combines two inverse processes: glycogenesis and glycogenolysis. During glycogenesis, glucose or glucose 6-phosphate molecules are converted into glycogen, using 1 adenosine triphosphate (ATP) and 1 ATP equivalent to add one glucose molecule to glycogen (Figure 3a). Only the glycolytic fibers store glycogen as a source of energy. Glycogen is tapped during times of nutritional insufficiency or rapid demand for muscle activity. This study cannot determine the direction of glycogen metabolism alone, only inferring potentially emphasized steps in either So or Ta. The glycogenesis-involved enzymes phosphoglucomutase (*Pgm2*, *Pgm2l1*) and glycogen synthase (*Gys2*) had increased expression in Ta (Figure 3a; Table 2). Phosphoglucomutase reversibly converts glucose 6-phosphate to glucose 1-phosphate. Increased expression of phosphoglucomutase could contribute to either glycogen metabolic processes.

Glycogen synthase is a critical enzyme in glycogenesis that converts glucose-1-phosphate into glycogen with the assistance of pyro-phosphorylase and a branching enzyme [25]. Increased expression of *Gys2* in Ta may imply that glycogenesis is the priority for this muscle (Figure 3a; Table 2). *Gys2* activity is regulated by dephosphorylation and phosphorylation by protein phosphatase 1 (PP1) and cyclic-adenosine monophosphate (AMP)-dependent protein kinase. PP1 is an enzyme complex with regulatory and catalytic subunits, of which two regulatory subunits, *Ppp1r1c* and *Ppp1r3g*, were expressed higher in So (Figure 3b; Table 2). *Ppp1r3g* regulates glycogen accumulation [26] and, along with the predicted inhibitory regulatory subunit *Ppp1r1c*, could be controlling glycogenesis in the So myofibers. The increased expression of the PP1 regulatory subunits in So could indicate a tighter control mechanism that prevents glycogen build-up. *Epm2a*, a factor that regulates PP1, was increased in Ta (Figure 3b; Table 2). *Epm2a* encodes for laforin, which interacts with PP1 and regulates the dephosphorylation of glycogen to promote glycogen branching [27]. Laforin is also involved in preventing cytotoxicity by protein ubiquitination [28], and it may control glycogen synthesis by ubiquitinating PP1, which would prevent the activation of glycogen synthase.

Glycogenesis must be inhibited to trigger glycogenolysis or glycogen catabolism. Glycogenolysis neither requires nor produces energy when breaking glycogen into glucose molecules. The process is triggered by the accumulation of gluconeogenic precursors, such as lactate or alanine. Glycogen synthase is inactivated to inhibit glycogenesis. The inactivation triggers the activation of phosphorylase kinase, which phosphorylates glycogen phosphorylase. Phosphorylase kinase is an enzyme complex of four subunits [29], three of which (*Phka1, Phkg1, Phkb*) also had increased expression in Ta (Figure 3c; Table 2). Phosphorylase kinase could be regulating the glycogenolysis in Ta without inhibiting glycogen synthase by ubiquitination. Glycogen phosphorylase pairs with a debranching enzyme to break off glucose 1-phosphate molecules from glycogen; a gene encoding for each (*Pygm* and *Agl*, respectively) was expressed higher in Ta (Figure 3a; Table 2).

### 3.4. Glucose Metabolism

Glucose metabolism consolidates glycolysis and gluconeogenesis enzymatic processes. The former process breaks glucose into an intermediate pyruvate, and the latter is the reverse process. Glycolysis converts glucose into pyruvate in 10 steps, consuming one glucose, two NAD+, two Pi, four ADP, and two ATP molecules to generate two pyruvate, two NADH, two H+, four ADP, and four ATP molecules. The net products are two pyruvate, two ATP, two NADH, and two H+ molecules. Gluconeogenesis converts two pyruvate, four ATP, two GTP, two NADH, two H+, and two H2O molecules into fructose 6-phosphate or glucose 6-phosphate. Overall, gluconeogenesis consumes six ATP, and glycolysis generates a net two ATP. Therefore, the whole glycolysis/gluconeogenesis cycle costs four ATP.

Three critical enzymes regulate the complete glycolysis process, and they are hexokinase, phosphofructokinase-1 (PFK1), and pyruvate kinase. Genes encoding for PFK1 (*Pfkm*) and pyruvate kinase (*Pkm*) had increased expression in Ta (Figure 4a; blue font, Table 2). PFK1 converts fructose 6-phosphate to F1,6-BP and is activated by high AMP concentrations and fructose 2,6-bisphosphate (F2,6-BP) [31]. Pyruvate kinase converts phosphoenolpyruvate to pyruvate when activated by high F1,6-BP concentration and is deactivated by an elevated ATP concentration. Four other isoenzymes involved in glycolysis and gluconeogenesis, aldolase (*Aldoa*, *Aldoc*), triosephosphate isomerase (*Tpi1*), phosphoglycerate kinase (*Pgk1*), and enolase (*Eno3* in Ta and *Eno2* in So), were increased in Ta (Figure 4a, Table 2).

The indirect regulator of glycolysis is the 6-phosphofructo-2-kinase/fructose 2,6-bisphosphatase (PFK2/FBPase2) complex. The PFK2/FBPase2 complex pulls fructose 6-phosphate out of the glycolytic process and converts it to F2,6-BP [30]. In Ta, three isoenzymes of PFK2/FBPase2, *Pfkfb3*, *Pfkfb1*, and *Pfkfb4* were increased (Figure 4b; blue font, Table 2).

When blood glucose concentrations are low, glucose can be generated through gluconeogenesis, an 11-step process for converting pyruvate to glucose. The enzymes pyruvate carboxylase, PEP carboxykinase, and fructose 1,6-bisphosphatase (FBPase1) are critical regulatory points in gluconeogenesis. FBPase1 (*Fbp2*) and PEP carboxykinase (*Pck1*) were increased in Ta and So, respectively (Figure 4a, Table 2). FBPase1 converts F1,6-BP to fructose 6-phosphate in the presence of a high concentration of AMP and F2,6-BP, while increased citrate levels deactivate it. PEP carboxykinase turns oxaloacetate into phosphoenolpyruvate and is enabled by a high ADP concentration.

All myofibers utilize glycolysis, whether or not they use aerobic and/or anaerobic respiration. Glycolysis occurs under both aerobic and anaerobic conditions because it does not require oxygen. Most of the DE genes associated with glucose metabolism had prominent expression in Ta compared to So. As a muscle that uses anaerobic metabolism to gain fuel, Ta may better utilize the anaerobic pathway due to the increased expression levels of glycolytic and regulatory enzymes.

### 3.5. Contraction

The Reactome pathway overrepresentation analysis identified contraction as another significant difference between So and Ta. The sliding filament theory explains the molecular basis of muscle contraction, where myosin and actin filaments interact in a particular manner to produce contractile force. Most contraction-related DE genes exposed were associated with the thin filament, thick filament, and Z-line of the sarcomere (Figure 5a, Table 3).

The thin filament is composed of actin subunits. *Actc1* had increased expression in Ta (Figure 5b, Table 3). Tropomyosin wraps outside the actin helix to stabilize the strand and rotate the thin filament to expose the binding sites. Two isoforms of tropomyosin were increased in So (*Tpm3*) [32] or Ta (*Tpm1*) [33] (Figure 5b, Table 3). The troponin complex is a regulatory element of the actin filament and is associated with calcium binding, inhibitory regulation, and tropomyosin binding. Tropomyosin-binding element encoding genes *Tnnt1* [34] and *Tnnt2* [35] were increased in So, while Ta had increased expression of *Tnnt3* [36] (Figure 5b, Table 3). The calcium and inhibitory troponin units, *Tnnc1*, *Tnni1*, and *Tnni3*, had an increased expression in So (Figure 5b, Table 3). The actin filaments are capped by tropomodulin *Tmod1,* which was expressed higher in Ta (Figure 5b, Table 3).

The thick filament consists of several strands of heavy myosin chains, which are two myosin heavy chains wrapped around one another with myosin heads at the ends that attach and flex to move along the thin filament. Myosin heavy chain isoforms unique to type I (*Myh7*) [1] and type II(B) (*Myh4*) [37] myofibers were increased in So and Ta, respectively (Figure 5c, Table 3).

Myosin light chains and myosin-binding proteins regulate myosin head movement. Myosin light chains are classified into essential (ELCs) and regulatory (RLCs). ELCs stabilize the myosin head, while RLCs stiffen the myosin neck domain. ELCs encoded by *Myl6b, Myl3*, and *Myl4* [2] and the RLCs encoded by *Myl2*, *Myl10* [38] were increased in So (Figure 5c, Table 3) while only *Mylpf* [39] was increased in Ta. *Mybpc2* [40] and *Mybph* that encode for myosin-binding proteins that assist in movement along the thin filament were increased in Ta (Figure 5c, Table 3).

The thin and thick filaments are held in an antiparallel configuration by a structure called the Z-line at the sarcomere ends (Figure 5d). The titin cap (*Tcap*) [41] is involved in mechano-electrical links between Z-lines and T-tubules and was expressed higher in So (Figure 5d, Table 3). Actinin (*Actn3*) [42] is involved in stabilizing the antiparallel formation and was increased in Ta (Figure 5d, Table 3). These structural differences could be altering the electrical activity and, in turn, the contractile properties of the different myofibers in the So and Ta muscles.

Several of the highest FC DE genes were part of the neuromuscular junction (NMJ) apparatus. Genes involved in the presynaptic membrane side included *Grp, Slc30a3, Asphd1, Nrxn3* [43], and *Nrsn2*, which were increased in So, and *Fam19a4* [44], *Kcnc4* [45], and *Fgfbp1* [46], which were increased in Ta (Figure 5e, Table 3). On the postsynaptic membrane side, *Tom1* [47], *Cntnap4* [48], *Wdr72* [49], and *Lypd1* were increased in So (Figure 5e; Table 3). Several of these genes are associated with endocytosis and neurotransmitter receptor signaling.

## 4. Discussion

Two broad-scale patterns emerged along with the three categories of overrepresented molecular pathways when examining the gene expression profiles of So and Ta muscles (Appendix A). First, DE genes involved in lipid, glucose, and glycogen metabolism pathways associated more with one muscle than the other (Table 1 and Table 2). Second, each muscle expressed unique genes encoding for similarly functioning isoenzymes, particularly for genes involved in the contraction and ion transport categories (Table 3 and Appendix A). We put these differences into context below, starting with the pathways that differentiate the muscle types. DE genes with the highest FC difference were not prevalent among genes associated with the overrepresented pathways. Therefore, DE genes with a greater than 20-fold expression difference were included for further investigation. Similarly seen in a previous study [54], we found that approximately 10% of DE transcripts were mainly related to fatty acid metabolism, structural components, and neuromuscular junction assembly.

### 4.1. Lipid Metabolism

FA lipids are an alternative energy source during fasting, starvation, and endurance exercise in oxidative muscles. Both So and Ta muscles contain oxidative myofibers, and both should express genes associated with FA metabolism (Figure 2). However, DE genes explicitly associated with FA transport and catabolism were increased in expression in So, which contains a higher percentage of oxidative fibers in its overall myofiber composition. In contrast, the 269-fold increased expression of *Cyp4f39* in Ta suggests that lipid catabolism to generate ceramides is more active in Ta. Ceramide accumulation has been linked to insulin resistance in type II diabetes [55]. The increased *Cyp4f39* expression may allow Ta to utilize ceramides to shift from being insulin-sensitive and from facilitating efficient glucose uptake [20] to becoming insulin-resistant. Alternatively, increased expression of *Cyp4f39* and presumed increased ceramide levels in Ta might be associated with differential apoptosis, cell cycling, or autophagy.

### 4.2. Glycogen and Glycosaminoglycan Metabolism

Glycogen acts as an energy reserve for myofibers by storing excess glucose molecules that are then utilized when blood glucose levels are low. Glycolytic or oxidative-glycolytic myofibers retain glycogen because they rely heavily on glucose as energy, and Ta has a high percentage of those myofibers; therefore, the increased expressions of the glycogen metabolic enzyme genes *Gys2*, *Pygm*, *Phka1*, *Phkg1*, and *Phkb* in Ta support the known metabolic composition of the muscle relative to So (Figure 3a; Table 2).

Hydrolysis of muscle glycogen to glucose occurs in lysosomes that engulf glycogen granules. The lysosome-associated DE genes, *Slc35d3*, *Atp6v0d2*, and *Galns*, exhibited increased expression in So, suggesting that lysosomal-related organelles in So and Ta myofibers perform slightly different functions (Appendix A). *Slc35d3* is associated with the biosynthesis of platelet-dense granules [56]. Platelet-dense granules contain high concentrations of calcium, adenine nucleotides, pyrophosphate, and polyphosphate molecules that enhance autophagy in lysosome-related organelles. *Atp6v0d2* is a part of vacuolar ATPases involved in proton translocation into vacuoles, lysosomes, or the Golgi apparatus to lower the pH [57]. The presence of *Atp6v0d2* suggested that lysosome-related organelles in So need to reduce their pH levels routinely. If lysosome-related organelles in So myofibers have an enhanced autophagy, the pH levels within those organelles would fluctuate as the cell engulfs and metabolizes more molecules from the extracellular space and may require proton pumps.

*Galns* is involved in glycosaminoglycan biosynthesis. Glycosaminoglycans are cell surface proteins with branches of sugars projecting into the extracellular matrix to support cell identity, adhesion, and growth. *Galns* encodes for a protein that breaks keratan sulfate off glycosaminoglycans [58] in lysosomes. *Stab2*, *B3gat1*, and *Cspg5* are also involved in glycosaminoglycan synthesis, indicating that So and Ta synthesize distinct glycosaminoglycans associated with cell adhesion and proliferation (Appendix A). *Stab2* is involved in the endocytosis of metabolic waste products, including circulating hyaluronic acid (HA) that promotes cell proliferation once the plasma concentration decreases. *B3gat1* is involved in generating HNK-1 carbohydrate (CD57) cell surface epitope associated with cell adhesion [59]. *Cspg5* is involved in chondroitin sulfate synthesis, another molecule added to proteoglycans and connected to cell adhesion, growth, migration, and receptor binding in the central nervous system (Appendix A). The high *Cspg5* expression in So suggests that neurons associated with the So muscle are marked differently.

### 4.3. Glucose Metabolism

Genes associated with glucose metabolism were increased in Ta more than So despite both muscles utilizing glycolysis. Yet, *Eno2* and *Pck1* were increased in expression in So compared to the 11 isoenzymes in Ta, perhaps due to the higher percentage of type II(B) glycolytic myofibers in Ta (Figure 4, Table 2). Glycolysis-related enzymes encoded by *Pfkm*, *Pkm Pfkfb3*, *Pfkfb1*, and *Pfkfb4* were increased in Ta (Figure 4b, Table 2). In contrast, two gluconeogenesis regulatory enzymes that regulate the reversion of pyruvate to glucose in skeletal muscle encoded by *Fbp2* and *Pck1* were expressed higher in Ta and So, respectively. Gluconeogenesis in Ta might not be as tightly controlled across several steps as glycolysis seems to be based on the 3-to-1 regulatory enzyme ratio. Gluconeogenesis may be controlled by the PFK2/FBPase2 complex instead of relying on the energy-costing enzyme PEP carboxylase seemingly utilized by So. *Pfkfb4* had a higher DE than *Pfkfb3*. *Pfkfb4* opposes *Pfkfb3* as it redirects glucose to the pentose phosphate pathway to promote the detoxification of reactive oxygen species and lipid and nucleotide biosynthesis. *Fbp2*, which was also increased in expression in Ta, converts F1,6-BP back into fructose 6-phosphate. These combined findings may indicate that Ta initiates gluconeogenesis at the PFK2/FBPase2 step of the glucose metabolism process to reduce energy loss.

### 4.4. Contraction

Several components that contribute to sarcomere formation were identified as DE between So and Ta, including MyHCs, myosin light chains, troponin subunits, and NMJ-associated proteins (Figure 5a–e). The presence of type I (*Myh7*) and type II(B) (*Myh4*)-associated MyHC isoforms among DE genes provided proof of the concept for this analysis. The fold change in differential expression of *Myh7* and *Myh4*, 34.7- and 57.7-fold, respectively (Table 3), reflected the appropriate percentage of types I and II(B) myofibers in each muscle. So and Ta both have type II(A) and II(D) myofibers, so differential expression of their specific MyHCs was predictably absent from our analysis. Our data also support a previous analysis correlating *Myh4* and *Myh7* with other genes associated with myofiber types I and II(B) [54], respectively (Table 3 and Appendix A). The myosin ELCs and RLCs mainly correlated to the appropriate muscle, except for *Myl4* and *Myl6* (Figure 5c, Table 3). The troponin complex subunits encoded by *Tnnt1*, *Tnni1*, and *Tnnc1* were unique and highly expressed in So (Figure 5b, Table 3), suggesting that the troponin complex in So might be modified relative to Ta to keep the actin filament in an open position longer, facilitating longer durations of contraction.

Factors affecting NMJ specification and calcium handling have been theorized to be among the non-myofibrillar, muscle-specific systems that allow for increased maximal isometric stress after P28 [60]. Synaptic modeling affects the motility or twitching of the skeletal muscle. Muscle identity is determined before innervation, yet innervation maintains muscle identity. During regeneration, only innervated So muscles can upregulate slow isoform mRNAs [61]. Yet, muscles do not completely change into another phenotype when neural cues are altered [62]. Intrinsic signaling from nerves and other sources, and environmental cues, dictate adult muscle phenotype [61]. Genes are associated with neurotransmission and synaptic vesicles [43] at the presynaptic side of NMJs, in addition to clathrin-mediated endocytosis [47,49], and cell adhesion [48] of the postsynaptic side was highly expressed in So. The release and retrieval of the synaptic vesicles and their contents may play an important role in the slow-twitch mechanism. On the contrary, Ta had increased expression of genes associated with the presynaptic side of NMJs only, which modulate neuronal excitability [44], broaden the action potential [45], and slow down the degeneration of NMJs [46]. These genes may explain how Ta and its associated neurons handle the stimulation necessary for rapid contraction. The DE genes related to the ion transport pathways may be associated with the postsynaptic side of NMJs in Ta myofibers, such as the voltage-dependent calcium channel subunits (Appendix A). However, all of this information and conjuncture is based on a small number of differentially expressed genes.

## 5. Conclusions

Two hindlimb muscles, So and Ta, which perform distinct locomotion functions, are characterized by unique metabolic and contractile properties. This focused investigation identified several genes uniquely tied to either So or Ta when these two muscle groups are compared. Approximately 10% of the mouse transcriptome was differentially expressed, including genes involved in lipid, glucose, and glycogen metabolism. Very highly differentially expressed genes such as *Cyp4f39* highlight previously undescribed potential differences in fatty acids between the SM that may underlie susceptibility to pathologies. The high DE of *Pfkm* and *Gys2* point to known differences in glucose and glycogen metabolism between the SM but add molecular details to the picture. Among contraction-related genes, RLCs, ELCs, and troponin complex subunits were higher expressed in So. DE of genes involved in NMJs suggests that electrochemical signaling is different between the two muscle groups, supporting the contractile differences.

Overall, this detailed study into the gene expression differences between these two muscle groups adds to the abundance of data on molecular differences among SM. Some similar results were found in single nuclei studies, which also identified unique expressions among individual skeletal muscle groups [54,63]. This study also highlights the continuing need to maintain and update gene and pathway ontology databases, especially adding more information on gene expression in different cell types.

## Figures and Tables

**Figure 1 genes-12-01718-f001:**
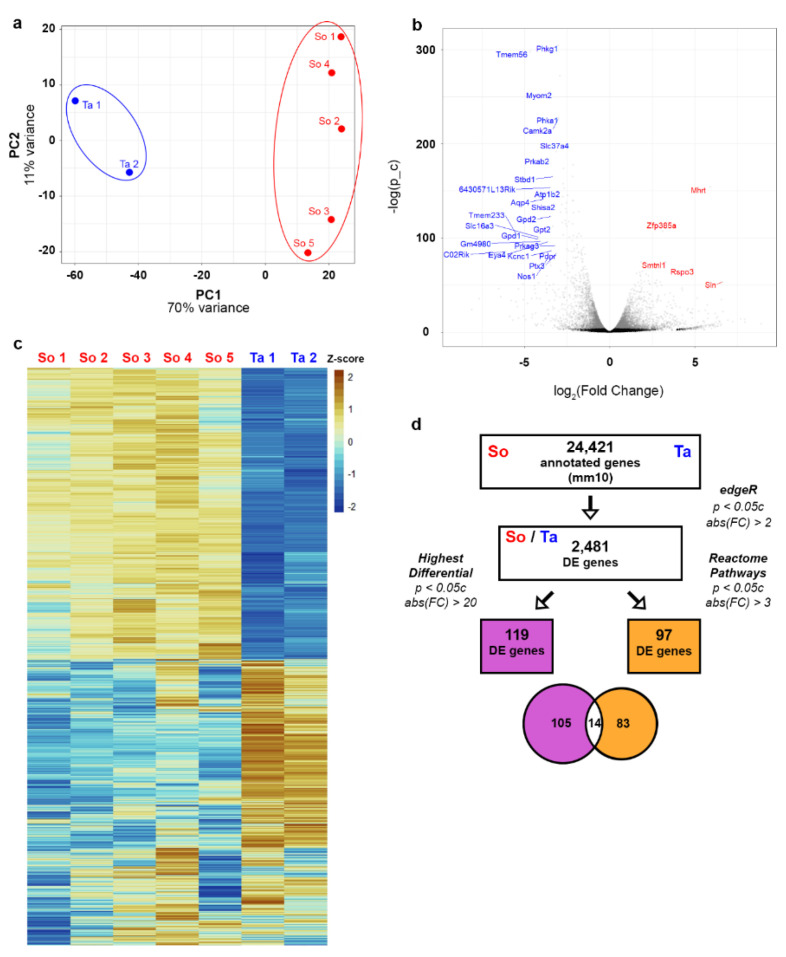
RNA-Seq workflow and data quality assessments. (**a**) Biplot of sample-standardized RNA counts (all genes) from the So (red) and Ta (blue) samples. Principal Component 1 (PC1) constitutes 70% of the variability among samples, and Principal Component 2 (PC2) constitutes 11% of the variability. (**b**) Volcano plot of all 24,421 annotated genes. Genes with an FDR-corrected *p*-value (p_c) less than 0.05 c are gray dots, and those with a *p*-value higher than 0.05 c are black. A select set of highly significant DE genes with large fold changes for So (log FC > 3, −log(p_c) > 50) are shown in red and for Ta (log FC < −3, −log(p_c) > 75) in blue. (**c**) Heatmap of z-scored log-transformed read counts. Positive z-scores are in orange, and negative z-scores are in blue. Each sample is a column, and each row represents a statistically significant DE gene from the 2481 set. (**d**) Workflow used to reduce the number of genes for further analysis. A subset of the 2481 DE genes with extremely high differential expression was selected; 119 DE genes had a log-fold expression greater than twenty (purple). Separately, 97 of the 2481 genes mapped to sixteen statistically overrepresented molecular pathways using the Reactome database R package (*p* < 0.05 c) and had a log-fold expression change >3 (orange).

**Figure 2 genes-12-01718-f002:**
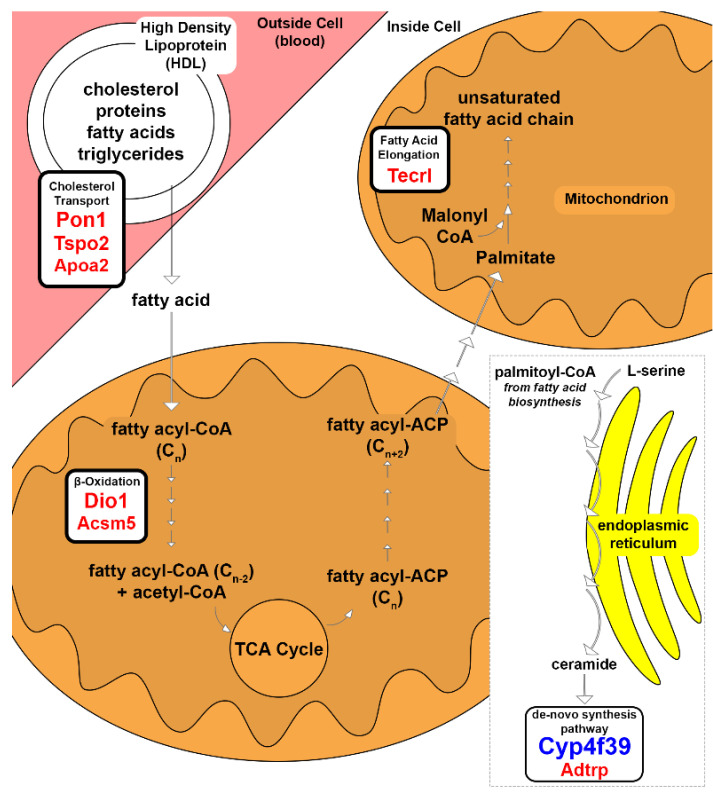
Lipid metabolism associated genes in So and Ta myofibers. Substrates and products of the enzymatic pathways for cholesterol transport, β-oxidation, FA elongation, and ceramide de-novo synthesis are shown based on their cellular location. The gene transcripts involved an enzymatic process and had increased expression in So or Ta, labeled in red or blue font, respectively. Font size correlates to relative FC.

**Figure 3 genes-12-01718-f003:**
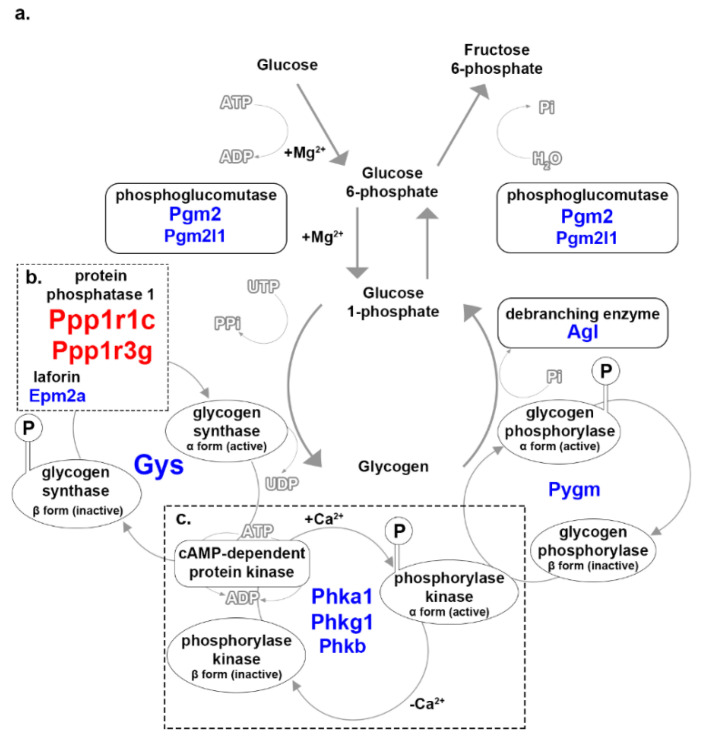
Glycogen metabolism associated genes in So and Ta myofibers. (**a**) Glucose is converted into glycogen during glycogenesis. Glycogenolysis converts glycogen back into glucose molecules. Represented are magnesium ions (+Mg^2+^), adenosine diphosphate (ADP), uridine-5′-triphosphate (UTP), pyrophosphate (PPi), and inorganic phosphate (Pi). (**b**) Two subunits of the protein phosphorylase 1 were increased in So. Represented are an encircled ‘P’ indicating phosphorylation and uridine-diphosphate (UDP). (**c**) Three isoform subunits of phosphorylase kinase were increased in Ta. Represented are the addition (+Ca^2+^) and the removal of calcium ions (−Ca^2+^). Substrates and products of the enzymatic steps in glycogen metabolism include the interconversion of inactive to active states of regulatory enzymes. Enzymes involved at specific stages are encircled and adjacent to the corresponding reaction arrow. Gene symbols located underneath enzyme names in red or blue font were increased in So or Ta, respectively. Font size correlates to relative FC.

**Figure 4 genes-12-01718-f004:**
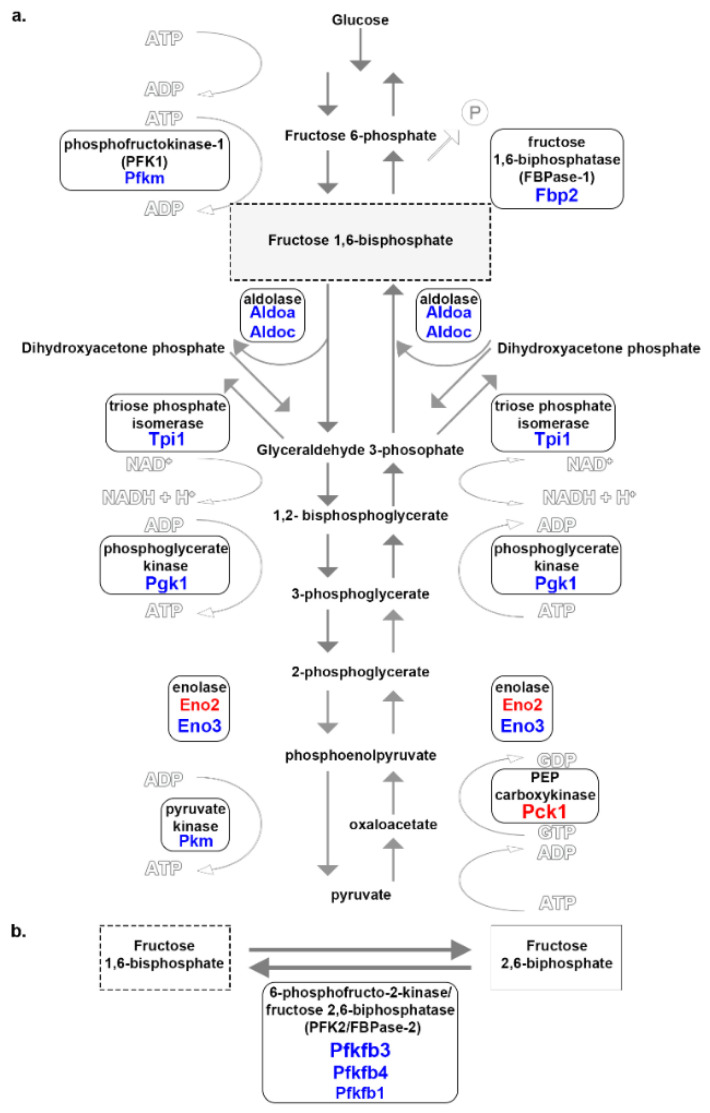
Glucose metabolism associated genes in So and Ta myofibers. Enzymes involved at specific stages are encircled and adjacent to the corresponding reaction arrow. (**a**) *Aldoa*, *Aldoc*, *Tpi1*, *Pgk1*, *Eno3*, *Pfkm*, *Pkm*, and *Fbp2* were increased in Ta. Isoforms of enolase (*Eno2*) and PEP carboxykinase (*Pck1*) were increased in So. Represented are guanosine triphosphate (GTP), guanosine diphosphate (GDP), and inorganic phosphate (P). (**b**) Subunits of the PFK2/FBPase2 complex, *Pfkfb3*, *Pfkfb1*, and *Pfkfb4*, were increased in Ta. The red or blue font of gene symbols underneath enzyme names represent increased expression in So or Ta, respectively. Font size correlates to relative FC.

**Figure 5 genes-12-01718-f005:**
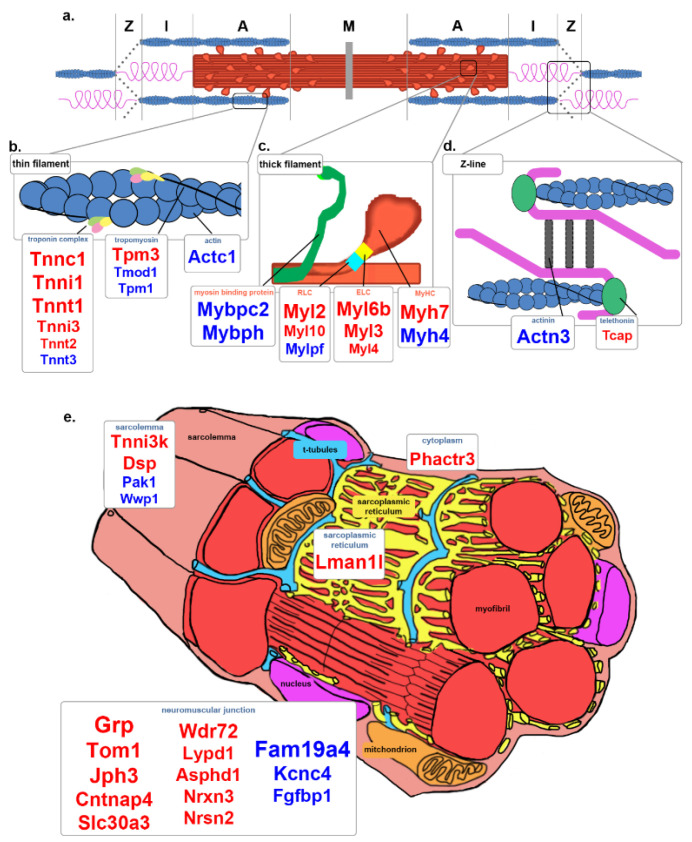
Contraction-related genes in So and Ta. (**a**) Graphic representation of the sarcomere. The M-line, A-zone, I-zone, and Z-line are depicted. The red band represents the thick myosin filament. The blue spheres represent the thin filament. Within the I-zone, resides an elastic protein called titin. (**b**) Graphic representation of thin filament structure. *Tpm3, Tnnc1, Tnni1, Tnnt1, Tnni3, Tnnt2*, and *Tnnt3* were increased in So. *Actc1, Tmod1, Tpm1*, and *Tnnt3* were increased in Ta. (**c**) Thick filament close-up. *Myh7, Myl6b, Myl3, Myl4, Myl2*, and *Myl10* were increased in So. *Myh4, Mylpf, Mybpc2*, and *Mybph* were increased in Ta. (**d**) Z-line close-up. The Z-line region of the sarcomere is where actin and titin are connected to an antiparallel complex, including titin cap (*Tcap*) and actinin (*Actn3*). (**e**) Myofiber structure. *Jph3, Lman1l, Phactr3, Tnni3k, Dsp, Grp, Tom1, Cntnap4, Slc30a3, Wdr72, Lypd1, Asphd1, Nrxn3*, and *Nrsn2* were increased in So. *Pak1, Wwp1, Fam19a4, Kcnc4*, and *Fgfbp1* were increased in Ta. Red or blue gene symbols underneath structure names represent increased expression in So or Ta, respectively. Font size correlates to relative FC.

**Table 1 genes-12-01718-t001:** DE genes associated with lipid, amino acid, and one-carbon metabolic pathways.

Gene Symbol	Gene Name	Fold Change (So/Ta)	Localization and Biochemical Properties
Metabolism			
*Dio1*	deiodinase, iodothyronine, type I	69.1	de-iodinate thyroid hormone interaction; FA oxidation; oxidative phosphorylation uncoupling; causes mitochondrial heat production [15]
*Pon1*	paraoxonase 1	68.83	protection against oxidation for HDLs and LDLs [10]
*Tecrl*	trans-2,3-enoyl-CoA reductase-like	55	FA elongation in polyunsaturated FA biosynthesis [17]
*Tspo2*	translocator protein 2	51.12	free cholesterol trafficking in erythroid cells [11]
*Acsm5*	acyl-CoA synthetase medium-chain family member 5	30.67	FA β-oxidation [13]
*Apoa2*	apolipoprotein A-II	27.64	modulator of reverse cholesterol transport [12]
*Akr1c18*	aldo-keto reductase family 1, member C18	24.68	catalyzes progesterone into 20-α-dihydroprogesterone (20-α-OHP) [21]
*Adtrp*	androgen dependent TFPI regulating protein	21.66	hydrolyzes FA esters of hydroxy FAs [22]
*Cyp4f39*	cytochrome P450, family 4, subfamily f, polypeptide 39	−269.23	1ω and 2ω diesters types and cholesteryl (O-Acyl)-w-hydroxy FAs (OAHFAs) production; FA ω-hydroxylase; acyl-ceramide production [19]
Amino Acid Metabolism			
*Aox3*	aldehyde oxidase 3	30.48	potentially linked to amino acid and retinol metabolisms; oxidizes aliphatic or aromatic aldehydes into carboxylic acid [23]
One-Carbon Metabolism			
*Gnmt*	glycine N-methyltransferase	26.87	Methylation of DNA, RNA, proteins, and lipids via the methionine cycle
Miscellaneous			
*Aldh1a7*	aldehyde dehydrogenase family 1, subfamily A7	−20.59	Oxidoreductase, NAD/NADP acceptor [24]

Positive or negative fold change represents increased gene expression in So or Ta, respectively.

**Table 2 genes-12-01718-t002:** Differentially expressed genes associated with glucose and glycogen metabolic pathways.

Gene Symbol	Gene Name	Fold Change (So/Ta)	Localization and Biochemical Properties
Glycogenolysis and Glycogenesis			
*Ppp1r1c*	protein phosphatase 1, regulatory inhibitor subunit 1C	75.78	indirect regulation of glycogenesis and glycogenolysis
*Ppp1r3g*	protein phosphatase 1, regulatory subunit 3G	36.99	PP1 control; indirect regulation of glycogenesis and glycogenolysis [25]
*Gys2*	glycogen synthase 2	−90.2	converts glucose 1-phosphate into glycogen [26]
*Phkg1*	phosphorylase kinase γ 1	−8.1	activates glycogen phosphorylase [29]
*Phka1*	phosphorylase kinase α 1	−8	activates glycogen phosphorylase [29]
*Phkb*	phosphorylase kinase β	−7.5	activates glycogen phosphorylase [29]
*Pgm2*	phosphoglucomutase 2	−6.7	converts of glucose 6-phosphate and glucose 1-phosphate
*Pygm*	muscle glycogen phosphorylase	−5.8	converts glycogen into glucose 1-phosphate
*Agl*	amylo-1,6-glucosidase, 4-α-glucanotransferase	−4.4	glucosidase activity that converts α-1,6-linked branches into α-1,4-linked branches
*Epm2a*	epilepsy, progressive myoclonic epilepsy, type 2 gene α	−3.5	facilitates the de-phosphorylation of glycogen to promote branching [27]
*Pgm2l1*	phosphoglucomutase 2-like 1	−3	conversion of glucose 6-phosphate and glucose 1-phosphate
Glycolysis and Gluconeogenesis			
*Pck1*	phosphoenolpyruvate carboxykinase 1, cytosolic	9.3	converts oxaloacetate to phosphoenolpyruvate
*Eno2*	enolase 2, γ neuronal	3	converts 3-PG into pyruvate
*Pfkfb3*	6-phosphofructo-2-kinase/fructose-2,6-bisphosphatase 3	−9.3	converts fructose-6-phosphate to F2,6-BP; indirectly regulates glycolytic flux [30]
*Eno3*	enolase 3, β muscle	−5.4	converts 3-PG into pyruvate in glycolysis
*Pgk1*	phosphoglycerate kinase 1	−5.1	converts 1,2-bisphosphoglycerate and 3-phosphoglycerate
*Tpi1*	triose-phosphate isomerase 1	−4.5	converts dihydroxyacetone phosphate to D-glyceraldehyde 3-phosphate
*Fbp2*	fructose bisphosphatase 2	−4.2	converts F1,6-BP to fructose 6-phosphate
*Pfkfb4*	6-phosphofructo-2-kinase/fructose-2,6-bisphosphatase 4	−4.1	converts F2,6-BP to fructose-6-phosphate; indirectly regulates glycolytic flux
*Pkm*	pyruvate kinase, muscle	−3.9	converts PEP to pyruvate
*Aldoc*	aldolase C, fructose-bisphosphate	−3.9	converts F1,6-BP to dihydroxyacetone phosphate
*Aldoa*	aldolase A, fructose-bisphosphate	−3.8	converts F1,6-BP to dihydroxyacetone phosphate; present in developing embryo and adult muscle
*Pfkfb1*	6-phosphofructo-2-kinase/fructose-2,6-bisphosphatase 1	−3.1	regulates glycolytic flux
*Pfkm*	phosphofructokinase, muscle	−3	regulates glycolytic flux; converts fructose 6-phosphate to F1,6-BP [31]

Positive or negative FC represents increased gene expression in So or Ta, respectively.

**Table 3 genes-12-01718-t003:** Differentially expressed genes associated with contractile related structures.

Gene Symbol	Gene Name	Fold Change (So/Ta)	Localization and Biochemical Properties
Neuromuscular Junction			
*Grp*	gastrin releasing peptide	149.61	modulates autonomic system; regulates male sexual function; conveys itch sensation
*Tom1*	target of myb1 trafficking protein	84.21	recruits clathrin to endosomal structures [47]
*Jph3*	junctophilin 3	66.32	formation of junctional membrane complexes; motor coordination [50]
*Cntnap4*	contactin associated protein-like 4	58.74	cell adhesion molecule and receptor [48]
*Slc30a3* *(Znt3)*	solute carrier family 30 (zinc transporter), member 3	34.8	transport of zinc in synaptic vesicles
*Wdr72*	WD repeat domain 72	33.63	endocytosis, protein reabsorption, and calcium excretion [49]
*Lypd1*	Ly6/Plaur domain containing 1	26.17	regulates neuronal nicotinic receptors
*Asphd1*	aspartate β-hydroxylase domain-containing protein 1	25.7	possibly neurotransmission and synaptic vesicle location/function
*Nrxn3*	neurexin III	24.74	synapse organization; regulating neurotransmitter release [43]
*Nrsn2*	neurensin 2	24.72	transporting small vesicles to perinuclear region to exit towards organelles
*Fam19a4 (Tafa4)*	TAFA chemokine like family member 4	−199.82	modulates neuronal excitability and controls somatic sensation threshold [44]
*Kcnc4 (Kv3.4)*	potassium voltage gated channel, Shaw-related subfamily, member 4	−33.68	broadens action potential to prevent somatic depolarization [45]
*Fgfbp1*	fibroblast growth factor binding protein 1	−28.02	slows neuromuscular junctions (NMJs) degeneration [46]
Sarcolemma, Sarcoplasm, and Sarcoplasmic Reticulum			
*Lman1l*	lectin, mannose-binding 1 like	61.63	binds to glycoproteins transported from endoplasmic reticulum (ER) to ER-Golgi intermediate [51]
*Tnni3k*	TNNI3 interacting kinase	27.58	promotes oxidative stress and myocyte death; part of costamere attached to sarcolemma [52]
*Phactr3*	phosphatase and actin regulator 3	26.09	binds to cytoplasmic actin and regulates PP1
*Dsp*	desmoplakin	20.92	assembles functional desmosomes; maintains cytoskeletal architecture [53]
*Pak1*	p21 (RAC1) activated kinase 1	−7.3	regulates muscle-specific kinase to maintain NMJs, actin remodeling, and glucose uptakes in skeletal muscle
*Wwp1*	WW domain containing E3 ubiquitin protein ligase 1	−3.5	cell proliferation and apoptosis
Thin Filament			
Actin			
*Actc1*	actin, α, cardiac muscle 1	−20.7	assembles into filaments that are involved in muscle contraction, cell motility, cell signaling, and vesicle movement; associated with fetal skeletal muscle
Tropomyosin and Tropomodulin			
*Tpm3*	tropomyosin 3, γ	29.17	controls the actin filament with tropomodulin [32]
*Tmod1*	tropomodulin 1	−3.6	controls the Ca^2+^-regulated thin filament end with tropomyosin
*Tpm1*	tropomyosin 1, α	−3.4	controls the Ca^2+^-regulated thin filament end with tropomodulin [33]
Troponin Complex			
*Tnnc1*	troponin C, cardiac/slow skeletal	47.7	binds to calcium and exposes the myosin head binding sites; slow skeletal muscle
*Tnni1*	troponin I, skeletal, slow 1	46.2	binds to actin and inhibits ATPase activity; slow skeletal muscle
*Tnnt1*	troponin T1, skeletal, slow	36.6	anchors to tropomodulin; slow skeletal muscle [34]
*Tnni3*	troponin I, cardiac 3	14.3	binds to actin and inhibits ATPase activity; cardiac muscle
*Tnnt2*	troponin T2, cardiac	3.3	anchors to tropomodulin; cardiac, embryonic, and neonatal skeletal muscles [35]
*Tnnt3*	troponin T3, skeletal, fast	−3	anchors to the tropomodulin; fast skeletal muscle [36]
Thick Filament			
Myosin Heavy Chains			
*Myh7*	myosin, heavy polypeptide 7, cardiac muscle, β	34.71	isoform of myosin present in slow (type I) skeletal muscle fibers [1]
*Myh4*	myosin, heavy polypeptide 4, skeletal muscle	−57.73	isoform of myosin present in adult type IIB skeletal muscle fibers [37]
Essential Myosin Light Chains			
*Myl6b*	myosin, light polypeptide 6B	54.1	controls cell adhesion, cell migration, tissue architecture, cargo transport, and endocytosis; promotes p53 protein ubiquitination and degradation
*Myl3*	myosin, light polypeptide 3	27	involved in force development and fine-scale coordinated muscle contraction
*Myl4*	myosin, light polypeptide 4	3.6	binds to Ca^2+^; embryonic skeletal muscle and atrial myocardium [2]
Regulatory Myosin Light Chains			
*Myl2*	myosin, light polypeptide 2, regulatory, cardiac, slow	41.7	stiffens myosin neck and aids in myosin head movement
*Myl10*	myosin, light chain 10, regulatory	4	stiffens myosin neck and aids in myosin head movement [38]
*Mylpf*	myosin light chain, phosphorylatable, fast skeletal muscle	−4	regulates myofilament activation by phosphorylation [39]
Myosin Binding Proteins			
*Mybpc2*	myosin binding protein C, fast type	−40.7	regulates of myofilament contraction [40]
*Mybph*	myosin binding protein H	−42.77	sarcomere contraction; maturation process of auto-phagosomal membranes; inhibition of non-muscle RLC MYL2A and MYH2A
Z-line			
*Tcap*	titin-cap	3.5	binds to titin in an anti-parallel complex and stabilizes Z-line [41]
*Actn3*	actinin α 3	−33.3	anchors actin filaments and supports sarcomere [42]

Positive or negative FC represents increased gene expression in So or Ta, respectively.

## Data Availability

FASTQ sequences were deposited to the NCBI gene expression omnibus (GEO) sequence read archive (SRA) using the accessions SRP127367 and SRP145066.

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
