# Peer review of "Gene Expression Profiling of Skeletal Muscles"

_genes, 2021, doi:10.3390/genes12111718_

Round 1

Reviewer 1 Report

The goal of the manuscript is to characterize gene expression of soleus and tibialis anterior, slow and fast skeletal muscles of the mouse. The authors obtained the expected result. The value of the study is diminished by unknown RNA to protein conversion factor, since at the end of the day, the number of proteins, expressed in muscle, matters.

Introduction: too general and unfocused, and the last paragraph is completely out of the line. Maybe the authors can make the Introduction fit the rest of the manuscript better. The sentence in line 58 misses a verb. The argument of using P30 mice is not convincing.

How many mice were used in the study? Please provide the number.

What are the SRP127367, NCBI SRA and SRP145066, NCBI SRA?

What is the “c” in 0.05c (line 106 for example)?

Define “FDR” (line 106)

Explain or reference the “Hypergeometric testing” (line 126)

It would be nice to introduce FC as the fold change somewhere in the text

Please explain the procedure of “manual examination” (line 133)

What are the PC1 and PC2 in figure 1a?

It is strange to see the fold change -20.7 for actin (Table 3). What does it mean? That actin expresses better in one muscle relative to the other?

Author Response

Review Report Form -- Reviewer #1

Comments and Suggestions for Authors

  1. The value of the study is diminished by unknown RNA to protein conversion factor, since at the end of the day, the number of proteins, expressed in muscle, matters.

The goal of this study was to conduct an in-depth analysis of the gene expression of two unique muscle types. Identification of differentially expressed transcripts provides information of presence/absence and quantification of a transcript, evaluation of the alternative/differential splicing events to predict protein isoforms and quantitative assessment of allele specific assessment. We disagree with the reviewer’s statement that “the number of proteins expressed in muscle matters” . First, a cell might have all proteins but some of them might be defected that will result in a phenotype, and thus is not a matter of number but a matter of quality, or a correct translational activity. Levels of gene expression is a reliable indicator of protein levels of expression. Second, proteome analysis provides different information and one approach does not overpower the other.

  1. Introduction: too general and unfocused, and the last paragraph is completely out of the line. Maybe the authors can make the Introduction fit the rest of the manuscript better.

The introduction was revised to reduce the generalization and refine the focus.

  1. The sentence in line 58 misses a verb.

We have corrected the sentence.

  1. The argument of using P30 mice is not convincing.

We have rewritten the introduction and include the rationale of using myofiber from postnatal day 30. This is the earliest time that single myofibers can be isolated from both muscle groups.

  1. How many mice were used in the study? Please provide the number.

Each sample (biopsy) corresponds to one mouse. Two mice for the tibialis anterior samples, and five for the soleus samples.

  1. What are the SRP127367, NCBI SRA and SRP145066, NCBI SRA?

The terms SRP127367 and SRP145066 are the reference numbers used by the National Center for Biotechnology Information (NCBI) Sequence Read Archive (SRA) that associates with the raw data files of the five soleus and two tibialis anterior myofiber biopsies used in this investigation. The following sentence was added to the manuscript (lines 85-88) to clarify this information: “The raw RNA-Seq data files for the five So and two Ta myofiber biopsies are publicly available through the National Center for Biotechnology Information (NCBI) Sequence Read Archive (SRA) under the reference numbers SRP127367 (So) and SRP145066 (Ta).”

  1. What is the “c” in 0.05c (line 106 for example)?

The “c” in 0.05c refers to the p-value being FDR-corrected. The definition was added to the text in lines 96-97 in the sentence: “The “c” in 0.05c will refer to the p-value being FDR-corrected, for the remainder of this manuscript.”

  1. Define “FDR” (line 106)

FDR stands for false discovery rate. The reference was added to the text in line 95 in the sentence: “Quantitative changes in gene expression levels between the So and Ta groups were discovered using the quasi-likelihood F test from the R package edgeR [9], which resulted in 6,123 differentially expressed (DE) genes (p < 0.05c, False Discovery Rate (FDR)-corrected) between So and Ta.”

  1. Explain or reference the “Hypergeometric testing” (line 126)

The hypergeometric testing is a part of the ReactomePA package. The text was rewritten to clarify this association on lines 206: “Hypergeometric testing provided by the ReactomePA R package identified 19 Reactome pathways as overrepresented with an adjusted p-value of less than 0.05c.”

  1. It would be nice to introduce FC as the fold change somewhere in the text

The original (FC) reference was accidentally removed and the authors would like to thank the reviewer for identifying this error. The abbreviation of FC is now located in the introduction, on line 69, in the sentence: “Additionally, DE transcripts with a high fold change (FC) not represented in the pathway analysis were further studied.”

  1. Please explain the procedure of “manual examination” (line 133)

Manual examination meant that the authors researched for each of the 105 highly FC genes and associated those genes into the categories of lipid metabolism, glycogen metabolism, glucose metabolism, and contraction. It was based on a deep literature search with the aim to identify their functions. Those genes were not placed in those groups when analyzing the data using the ReactomePA R package.

  1. What are the PC1 and PC2 in figure 1a?

PC1 represents Principal Component 1, in which 70% of the variability among the samples exists. PC2 represents Principal Component 2, in which 11% of the variability among the samples exists. The following sentence was added to the Figure 1 legend to clarify this information in lines 222-223: “Principal Component 1 (PC1) constitutes 70% of the variability among samples, and Principal Component 2 (PC2) constitutes 11% of the variability among samples.”

  1. It is strange to see the fold change -20.7 for actin (Table 3). What does it mean? That actin expresses better in one muscle relative to the other?

The negative (-20.7) value means that the Actc1 transcript was expressed 20.7 times lower in soleus myofiber when compared to tibialis anterior myofiber.

Reviewer 2 Report

This manuscript provided the information of gene expression in soleus and tibia and showed the differences between them. I have several comments as below.

  1. In figure 1d, authors provided the workflow to show how to select the critical genes and pathways, but I don’t understand why to indicate “Fourteen genes were shared between the highly DE (119) and pathway associated (97) genes.”, these fourteen genes have special functions or other meanings?
  2. The rationales for choosing and studying soleus and tibia are not clear. Authors didn’t provide the enough conclusion to demonstrate the novelty of this manuscript.

Author Response

Review Report Form -- Reviewer #2

Comments and Suggestions for Authors

  1. In figure 1d, authors provided the workflow to show how to select the critical genes and pathways, but I don’t understand why to indicate “Fourteen genes were shared between the highly DE (119) and pathway associated (97) genes.”, these fourteen genes have special functions or other meanings?

These fourteen genes do not have special functions or other meanings. The authors wanted to clarify that the two separate analyses had overlapping results.

2.The rationales for choosing and studying soleus and tibia are not clear.

The authors would like to apologize for the lack of clarity on the rationale behind studying soleus and tibialis anterior. These two skeletal muscles represent two major classes of myofiber composition , which in turn represent two major pathways of energy production, ß-oxidation and glycolysis respectively. The introduction was revised to better address the reviewer’s concern.

3.Authors didn’t provide the enough conclusion to demonstrate the novelty of this manuscript.

We have modified the discussion and we have highlighted the conclusions of the study.

Round 2

Reviewer 2 Report

As previous revision, I had a question as "In figure 1d, authors provided the workflow to show how to select the critical genes and pathways, but I don’t understand why to indicate “Fourteen genes were shared between the highly DE (119) and pathway associated (97) genes.”, these fourteen genes have special functions or other meanings?" Authors answered "These fourteen genes do not have special functions or other meanings. The authors wanted to clarify that the two separate analyses had overlapping results."

Why need to clarify these two separate analyses had ovelapping results? If the overlapping results (14 genes) have no special issue the authors want to point out.